# Cultivating clean sport environment with athlete support personnel (ASP): A study on anti-doping knowledge, attitudes, and practices of ASP

**Ming Chiang Lim**[1,2☯], **Gobinathan Nair**[3☯], **Eng Wee Chua**[4☯], **Tuan Mazlelaa Tuan Mahmood**[1☯], **Farrah-Hani Imran**[5‡], **Ahmad Fuad Shamsuddin**[6‡], **Adliah Mhd Ali**[1]*

1 Faculty of Pharmacy, Centre for Quality Management of Medicines, Universiti Kebangsaan Malaysia, Kuala Lumpur, Malaysia, 2 Department of Pharmacy, Hospital Sultan Haji Ahmad Shah, Temerloh, Pahang, Malaysia, 3 Southeast Asia Regional Anti-Doping Organisation, Singapore Sports Council, Singapore, Singapore, 4 Faculty of Pharmacy, Centre for Drug and Herbal Development, Universiti Kebangsaan Malaysia, Kuala Lumpur, Malaysia, 5 Faculty of Medicine, Universiti Kebangsaan Malaysia, Cheras, Kuala Lumpur, Malaysia, 6 Faculty of Pharmacy and Health Sciences, Universiti Kuala Lumpur Royal College of Medicine Perak, Ipoh, Perak, Malaysia

☯ These authors contributed equally to this work.
‡ These authors also contributed equally to this work
* adliah@ukm.edu.my

**Data Availability Statement:** "All relevant data are within the paper and its Supporting Information files."

## Abstract

Athlete support personnel (ASP) work closely with, treat, or assist an athlete participating in or preparing for sports competition. Their involvement in preventing and eliminating doping is crucial. This study investigated the knowledge, attitudes, and practices related to doping in sports among ASP from Southeast Asian countries. An anonymized self-administered questionnaire assessing knowledge, attitudes, and practices related to doping in sports issues was administered to ASP from Southeast Asian countries. Overall, 596 respondents from eleven countries participated in the study. The majority were male (67.1%), non-health-care professionals (89.4%), and retired elite athletes (57.7%). Their knowledge was found to be poor, reflected in a mean score of 16.1±5.4 out of 30. Attitudes towards doping, as measured by the Performance Enhancement Attitude Scale (PEAS), scored 18.1±9.4, indicating a negative attitude. While some respondents provided information on medication and supplements use in sports to athletes, only 11.8% reported regular updates on doping in sports topics. Meanwhile, the knowledge and PEAS scores were significantly different between the genders (p = 0.04; p = 0.02). The knowledge score was also negatively correlated with the PEAS (p<0.01). This study highlights significant knowledge gaps among ASP in Southeast Asia regarding anti-doping practices. Enhancing their knowledge and fostering positive attitudes toward anti-doping efforts can promote a culture of doping-free sports, particularly among the emerging generation of young athletes they support.

**Funding:** Ming Chiang, Gobinathan Nair, Eng Wee, Tuan Mazlelaa, Ahmad Fuad, Adliah Social Science Research Grant Program (2021 A-9, NF-2023-002) World Anti-Doping Agency https://www.wada-ama.org/en Approval to publish was received from the funder, no amendment was done by the funder on the study protocol and final manuscript. The funders had no role in study design, data collection and analysis, decision to publish, or preparation of the manuscript.

**Competing interests:** The authors have declared that no competing interests exist.

# Introduction

Athletes, as they progress in their professional sports careers, harbour a constant desire to improve and succeed. In their pursuit of a successful sports career, athletes are never alone. They are surrounded by athlete support personnel (ASP), who assist them in various facets of their athletic endeavours, including training routines, injury prevention and treatment, and daily activities. Athlete support personnel (ASP) are defined by the World Anti-Doping Agency (WADA) as "any coach, trainer, manager, agent, team staff, official, medical, paramedical personnel, parent or any other person working with, treating or assisting an athlete participating in or preparing for sports competition" [1]. ASP are often considered as the most dependable and trustworthy individuals in athletes' careers, as athletes spend a substantial amount of time with them. Coaches, in particular, were found to be the most important sources of information on supplement use among young athletes [2]. Moreover, studies suggest that adolescent athletes are more likely to seek information on performance-enhancing substances and may be swayed by their coaches' behaviour towards doping, if the coach were to verbally persuade them [3,4].

Past literatures highlight the impact of ASP on athletes' decisions with considerable evidence that they are poorly informed on doping issues. Blank et al. (2013) pointed out that most parents of Austrian junior athletes have shown an insufficient knowledge on anti-doping topics [5]. Similarly, coaches, physical trainers, and technical staff have been reported not knowing the meaning of WADA and the substances listed in the WADA Prohibited List [6]. Despite being the closest companions and primary confidants of athletes, ASP have reported a lack of engagement with, or opportunities to engage with formal anti-doping education [6]. This may lead to lack of confidence among ASP to discuss and disseminate anti-doping education to athletes. A recently published systematic review suggested that although coaches are generally aware of the importance of anti-doping, they had limited knowledge regarding prohibited substances and the consequences of non-compliance to doping control. Despite this, many still provide anti-doping advice without referring to the proper references or undergoing formal anti-doping education [7].

In light of the inadequate knowledge, previous works in this field indicated that both healthcare professionals (HCP) and non-healthcare professionals generally exhibit negative attitudes towards doping in sports [6,8,9]. Most physicians view the use of banned substances by athletes as unethical and believe that healthcare professionals should actively combat the misuse of substances in sports [10]. Majority of the pharmacists also acknowledge the importance of doping prevention initiatives, and they could play a vital role in doping prevention, yet they lack the confidence to discuss and disseminate anti-doping education to athletes [8,11]. Furthermore, while most coaches have been found to recognise their role in anti-doping initiatives, some prioritize maximizing athletes' performance over anti-doping [7]. However, there is no excuse for ignorance when it comes to their roles against doping. The World Anti-Doping Code (WADC) clearly defines the responsibility of the ASP under Article 21.2, stating that all support person should be aware of and comply with anti-doping responsibilities, cooperate with athlete testing programme, and use their influence to foster anti-doping attitudes in athletes [1]. ASP working closely with young athletes should undergo proper training because attitudes towards doping are shaped early in athletes' careers [12]. These young athletes may either continue their careers as elite athletes or transition to supporting roles. Therefore, comprehensive education should be provided to ASP working with youth athletes to instil ethical values and promote healthy behaviours early in their careers [13].

Early anti-doping education is important, as doping violation can lead to serious consequences, including sanctions, reputational damage, and career-ending penalties [1]. Cases of

athletes being suspended for violating anti-doping regulations have been documented in Southeast Asia, and the increasing numbers of cases involving younger athletes is worrying [14–16]. These instances highlight the needs for comprehensive anti-doping education among ASP to ensure that athletes are adequately educated and supervised by their closest career companion and protected against doping. While research on anti-doping has been conducted globally, studies in Southeast Asian countries remain limited. Given the collectivistic and interdependent culture, the influence of ASP on athletes maybe even more prominent in these regions [17]. Thus, this study aimed to conduct survey to enhance our understanding of the knowledge, attitude, and practices (KAP) of ASP working with elite youth athletes from Southeast Asia regarding doping in sports.

## Materials and methods

### Study design

This is a cross-sectional study targeting the ASP from national sports schools or elite youth training programmes in Southeast Asian countries regarding issues related to doping in sports. The study was conducted between 1st January and 31st December 2023. Invitations to participate in the study were emailed to all Member Country Representatives from Southeast Asia Regional Anti-Doping Organisation (SEARADO) member countries comprising Brunei Darussalam, Cambodia, Indonesia, Lao People's Democratic Republic, Malaysia, Myanmar, Philippines, Singapore, Thailand, Timor-Leste, and Vietnam. Forty-six sports schools and youth training programme from these countries were selected based on their active involvement in national and international sports events or recommendations from the representatives of respective National Anti-Doping Organizations (NADOs), aiming to reflect the elite sports community accurately. In this study, ASP were defined based on the WADC as "individuals working with, treating or assisting an athlete participating in or preparing for sports competition" [1]. The definition included, but was not limited to, chiropractors, coaches, dietitians, team manager, medical assistant, nursing profession, nutritionists, occupational therapist, pharmacists, physicians, physiotherapists, psychologists, sport administrator, and sports trainers. Academic teachers who solely focus on academic subjects and are not directly involved in preparing the athletes for sports competition were excluded. The study did not include parents of youth athletes. They can be one of the most influential ASP, but our focus was on ASP who had a professional relationship with the athletes and more clearly delineated roles and responsibilities related to their training and support. By focusing on ASP working within educational facilities, we could gain more consistent and specific insights relevant to the study's objectives. Participants were also excluded if they were not able to read or understand either English or their national language.

### Design of a self-administered questionnaire

The respondents' knowledge, attitude, and practice on doping in sports were assessed using a self-administered questionnaire, which comprised four sections. Section A collected demographic information, including age, gender, educational level, type of sport-related profession, and years of practice in current supporting role. Questions about the specific types of sports where the ASP provided support were not included, as the ASP might work across several sports, and specifying the sport could reveal individual identities.

Section B assessed knowledge regarding doping, with questions adapted from previous literature and revised according to the latest WADA Prohibited List 2023 and the WADC [9,18,19]. Section B was also tailored to the respective roles for the HCP and the non-HCP, as the depth of the information they need to deliver to athletes differ. Knowledge assessment was

based on the marks obtained by respondents, with each correct answer earning one mark, while incorrect answers or responses of "Don't know" receiving zero marks. Thus, the maximum total score for this section was 30. Total scores ≥84% were classified as "good", scores between 61% and 83% were rated as "moderate", and scores ≤60% were classified as "poor" [18]. An explanatory statement was included at the beginning of the section that advises the respondents not to refer to any resources when answering the knowledge-based questions.

Section C measured the attitude of ASP towards doping using the Performance Enhancing Attitude Scale (PEAS) [20,21]. The PEAS is a unidimensional and reliable 8-item instruments scale measured on a 6-point Likert-type scale ranging from strongly disagree (1) to strongly agree (6). Responses were categorized as positive (slightly agree to strongly agree) or negative (slightly disagree to strongly disagree). The shorter 8-item version we used in this study was reported to have a good model fit and a higher Cronbach' alpha value than the original 17-item version [21]. In the final section, respondents were asked about their practices and experiences regarding anti-doping activities. This allowed us to estimate the frequency of enquiries from young athletes regarding medication and supplements use in sports, as well as their past experiences with doping in sports training.

The questionnaire was prepared in English and translated by professional editors and translators into Malay, Indonesian, Lao, Khmer, Burmese, Tagalog, Thai, Tetum, and Vietnamese. It was then back-translated by the respective NADO's educational teams to ensure the phrasing and terminology used in the questionnaire were appropriate and suitable for the context of doping in sports. Ethical approval was obtained from the Research Ethics Committee, Universiti Kebangsaan Malaysian (UKM PPI/111/8/JEP-2022-405), the Education Policy Planning and Research Division of Ministry of Education Malaysia (KPM.600-3/2/3-eras(12752), and the Sports, Co-curricular and Arts Division of Ministry of Education Malaysia (KPM.600-2/1/4 Jld. 6 (56).

## Psychometric evaluations and validation

Prior to the distribution of the questionnaire, it was pre-tested by several validation tests and psychometric evaluations. A content validity test was conducted with eight independent reviewers from academic institutions and anti-doping agencies not involved in developing the questionnaire to ensure the concepts of interest were comprehensively represented by the items in the questionnaire. The invited experts were briefed on the study, along with a critical appraisal sheet with the following four inquiries: 1) the relevance of each question; 2) the importance of each question; 3) the clarity of each question; and 4) additional comments or recommendations for improvement of each section.

Item-Content Validity Index (I-CVI) and Scale-level Content Validity Index (S-CVI) were used to rate the relevance of the item. The reviewers assessed the relevance of each question based on a 4-point ordinal Likert scale, where "4" (highly relevant) and "3" (quite relevant) indicated agreement with a statement, while "2" (somewhat relevant) and "1" (not relevant) denoted disagreement. The I-CVI for the questionnaire ranged from 0.88 (7 agreed) to 1.00 (8 agreed), which represented high content validity for all items; while the Scale-level Content Validity Index based on the average method (S-CVI/Ave) for the questionnaire was 0.97, which represented high content validity overall. These computed CVIs were considered satisfactory [22].

On the other hand, Content Validity Ratio (CVR) was applied to quantify the necessity of the items. The reviewers rated the importance of each question based on a 4-point ordinal Likert scale, which consisted of "4" (essential), "3" (useful but not essential), "2" (provide some information but not essential), and "1" (not essential). The CVR of the questionnaire

ranged from 0.75 (7 agreed) to 1.0 (8 agreed), which showed that most of the panellists agreed on the necessity of the items in the questionnaire; so, none of the items were removed [23]. Minor grammatical adjustments were made based on the reviewers' comments to improve the clarity of the questionnaire without changing the original meaning of the questions.

After the content validation was completed, a pilot test was carried out with 30 ASP from five sport schools in Malaysia. For the face validity which gathered respondents' feedback on the clarity of the questionnaire, no major comment was received, thus all items were maintained. Besides, internal consistency was measured using reliability analysis. The Cronbach's alpha value was found to be 0.87, which surpassed the acceptable threshold for reliability [24]. For the knowledge section of the questionnaire, the item difficulty index (IDI) was calculated for each question. Most of the items had difficulty indexes between 0.2 and 0.8, indicating that their difficulty levels were neither too easy nor too difficult [25]. Only one question about the status of insulin as prohibited substance had a difficulty index of 0.13, as most of the respondents were not aware that insulin is not allowed in sports.

After the pilot test, the questionnaire was distributed among the ASP in the selected institutions. ASP were approached with assistance from the representatives of their respective NADOs. They received emails or WhatsApp messages containing an overview of the study and a Google Form link to the questionnaire in both English and their native language. ASP who agreed to participate were asked to provide written informed consent and complete the survey. The consent form was available on the first page of the questionnaire, and no sign-in was required to ensure anonymity. Once the data collection period ended, the link was disabled, and only researchers could access it. The data collected were exported from the Google Form and stored in a separate offline document on a password-protected computer, and all data stored in Google Cloud were deleted. For institutions requesting physical copies of the bilingual questionnaire, these were distributed along with informed consent forms during outreach programs. Respondents' personal information was removed from the consent form, and the questionnaire data were entered into an Excel spreadsheet separately by researchers for confidentiality.

## Data analysis

Data were analysed using SPSS software version 23. Categorical demographic data were presented in frequencies and percentages, with statistical significance set at p-value <0.05 for all inferential tests. The distribution of the numerical data in the study was examined using the Kolmogorov-Smirnov test, revealing that the age of the respondents, the year(s) of practice as supportive person, and overall knowledge score were normally distributed (p>0.05). For the analysis of relationship between respondents' knowledge score with demographic data, independent t-test tests were used to look for association of knowledge score with gender, health professional status, ex-athlete status while Pearson correlation test was used to associate knowledge score with years of practice as support personnel. Meanwhile, the association between respondents' PEAS score with demographic data and knowledge were examined. Demographic data including gender and status of being an elite athlete were associated with PEAS score using independent t-test. One-way Analysis of Variance (ANOVA) test was used for examining the relationship between knowledge grade and PEAS score. Lastly, to evaluate the relationship between respondents' experience and practice with demographic data, knowledge and attitude, Pearson chi-square test was used for categorical data while independent t-test was used for the relationship between experience with the knowledge score and PEAS score.

## Results

### Demographic characteristics of participants

A total of 596 respondents from the invited SEARADO member countries completed the questionnaire. The majority were male (n = 400, 67.1%), with a mean age of 38.4 years and standard deviation (SD) of 9.4. Most of them held a bachelor's degree (n = 342, 57.4%), were non-healthcare professionals (n = 533, 89.4%) and had previous experience as athletes participating in national or international sports events (n = 344, 57.7%). Among the respondents, the majority were coaches (n = 352, 59.1%), followed by sport trainers (n = 58, 9.7%), sport administrators (n = 55, 9.2%) and referees (n = 23, 3.9%). Among the invited countries, Vietnam contributed the highest number of respondents (n = 172, 28.9%), followed by Indonesia (n = 143, 24.0%) and Philippines (n = 93, 15.6%). On average, respondents had worked in their current role for 9.7 years (SD = 7.5). The demographic characteristics of the respondents are summarized in Table 1.

### Knowledge on doping in sports

The mean knowledge score for the respondents was reported to be 16.1 (SD = 5.4), indicating a poor to moderate level of knowledge related to drugs in sports. Detailed analysis of the six knowledge domains is presented in Table 2. While respondents were generally aware that anabolic-androgenic steroids (n = 302, 50.7%), stimulants (n = 332, 55.7%), and morphine (n = 325, 54.5%) are prohibited in sports, awareness was lower for insulin (n = 130, 21.8%) and beta-2 agonists (n = 164, 27.5%). Besides, most of the respondents were able to answer correctly for the descriptions about therapeutic use exemption (n = 342, 57.4%) and but most were unsure about what athlete biological passport is (n = 310, 52.0%). Most of the respondents were aware that doping violations include the attempted use of a prohibited substance or method by athlete (n = 471, 79.0%), the attempted administration of a prohibited substance or method to an athlete by an athlete support person (n = 397, 66.6%), and failure to submit to sample collection (n = 332 55.7%). They were also able to identify their respective NADO correctly (n = 349, 58.6%) and understood the roles of the WADA, which include coordinating anti-doping activities, maintaining the WADC and the Prohibited List. However, almost half of the respondents were unaware that accreditation and reaccreditation of laboratories for testing procedures were part of WADA roles. Regarding their role as athlete support personnel, most recognized the importance of adhering to anti-doping policies, rules, and cooperating with athlete-testing programs. However, fewer acknowledged the role of influencing athlete values and behaviour to foster anti-doping attitudes (n = 302, 50.7%).

For the analysis of the relationship between the respondents' knowledge score with demographic data, knowledge score was found to be significant different between gender (t(595) = -2.03, p = 0.04), with females scoring slightly higher (16.6±5.0) than males (15.6±5.4). However, knowledge scores did not significantly differ by professional status (p = 0.47) or previous athletic experience (p = 0.42). The knowledge score was also not correlated with their years of experience as athlete support personnel (p = 0.13).

### Attitude towards doping in sports

The mean score for the performance enhancing attitude scale (PEAS) was 18.1 with SD of 9.4, indicating a negative attitude toward drugs use in sports. Table 3 summarizes the PEAS in detail. There was a significant difference between the attitude and the knowledge grade (F (2,593), p = 0.01). Post-hoc analysis revealed a statistically significant difference for the mean PEAS for participants with moderate knowledge (mean PEAS of 16.94, p = 0.02) compared to

**Table 1. The respondents' demographic characteristics (n = 596).**

| Demographic variables | Number (n) | Percentage (%) |
|---|---|---|
| **Gender** | | |
| Male | 400 | 67.1 |
| Female | 196 | 32.9 |
| | | |
| **Highest education level** | | |
| Primary education | 2 | 0.3 |
| Secondary education | 71 | 11.9 |
| Diploma | 36 | 6.0 |
| Bachelor's degree | 342 | 57.4 |
| Master's degree | 127 | 21.3 |
| Doctorate' degree (PhD) | 18 | 3.0 |
| | | |
| **Country currently serving** | | |
| Brunei | 13 | 2.2 |
| Cambodia | 16 | 2.7 |
| Indonesia | 143 | 24.0 |
| Laos | 12 | 2.0 |
| Malaysia | 26 | 4.4 |
| Myanmar | 34 | 5.7 |
| Philippines | 93 | 15.6 |
| Singapore | 17 | 2.9 |
| Thailand | 62 | 10.4 |
| Timor-Leste | 8 | 1.3 |
| Vietnam | 172 | 28.9 |
| | | |
| **Professional status** | | |
| Chiropractors | 0 | 0 |
| Dietitian | 5 | 0.8 |
| General practitioner/ medical doctor | 16 | 2.7 |
| Medical assistant | 1 | 0.2 |
| Nursing profession | 14 | 2.3 |
| Nutritionist | 3 | 0.5 |
| Occupational therapist | 1 | 0.2 |
| Pharmacist | 2 | 0.3 |
| Physiotherapist | 10 | 1.7 |
| Psychologist | 9 | 1.5 |
| Coach | 352 | 59.1 |
| Manager | 18 | 3.0 |
| Sport administrator | 55 | 9.2 |
| Sport trainer | 58 | 9.7 |
| Others | 52 | 8.7 |

poor knowledge (mean PEAS of 19.29) but not with good knowledge (mean PEAS of 16.39). There was no statistically significant difference of the PEAS between participants with good and moderate knowledge (p = 0.95). Gender differences were also observed, with males exhibiting significantly higher PEAS scores (19.17±9.85) than females (17.18±8.58, t(595) = 2.41, p = 0.02). Attitude did not significantly differ based on previous athletic participation (p = 0.16) or attendance at anti-doping courses (p = 0.87). A weak, negative correlation was observed between knowledge and PEAS scores (r(594) = -0.2, p<0.01).

## Practice and experience in drugs use in sports

The current study showed that more than 55% of the respondents had experience in providing information to the athletes about medications and dietary supplements use in sports. However, it is worrying that 45.6% of the respondents claimed that they had counselled athletes about

**Table 2. The respondents' knowledge of doping in sports (n = 596).**

| Domains/ Variables | Correct answer | Number of correct answers, n (%) | Number of wrong answers, n (%) |
|---|---|---|---|
| **Knowledge on prohibited substances in sports**<br>1. The substances classified by the World Anti-Doping Agency (WADA) as prohibited in sports under the WADA Prohibited List 2023 include/ You may want to alert your athletes if they are taking the following substance(s) as they could be prohibited in sports under the WADA Prohibited List 2023: | True | 302 (50.7) | 294 (49.3) |
| | True | 224 (37.6) | 372 (62.4) |
| | True | 210 (35.2) | 386 (64.8) |
| | True | 164 (27.5) | 432 (72.5) |
| (i) Anabolic androgenic steroids (AAS) | True | 130 (21.8) | 466 (78.2) |
| (ii) Peptide hormones | True | 332 (55.7) | 264 (44.3) |
| (iii) Growth factors | True | 271 (45.5) | 325 (54,5) |
| (iv) Beta-2 agonists/ Reliever inhaler for asthma attack | False | 328 (55.0) | 268 (45.0) |
| (v) Insulin | True | 325 (54.5) | 271 (45.5) |
| (vi) Stimulants | True | 180 (30.2) | 416 (69.6) |
| (vii) Diuretics/ Water pills | False | 370 (62.1) | 226 (37.9) |
| (viii) Nicotine/ Cigarettes | False | 284 (47.7) | 312 (52.3) |
| (ix) Morphine | True | 264 (44.3) | 332 (55.7) |
| (x) Beta-blockers/ Medication for high blood pressure | | | |
| (xi) Caffeine | | | |
| (xii) Alcohol | | | |
| 2. Some athletes use diuretics as masking agents to hide the presence of other banned substances in their urine/ Some athletes use water pills to increase urine volume to hide the presence of other banned substances in their urine | | | |
| **Knowledge on National Anti-Doping Agency**<br>1. What is the title of national anti-doping agency in your country? | Answer based on country | 349 (58.6) | 247 (41.4) |
| **Knowledge on Therapeutic Use Exemption & Athlete Biological Passport**<br>1. Therapeutic Use Exemption (TUE) allows athletes to use prohibited substances for medical reasons in or out of competition | True | 342 (57.4) | 254 (42.6) |
| 2. The Athlete Biological Passport (ABP) programme monitors specific parameters in the body to measure the effects of doping without detecting the doping substance or method. | True | 286 (48.0) | 310 (52.0) |
| **Knowledge on roles of World Anti-Doping Agency (WADA)**<br>1. The roles of the World Anti-Doping Agency (WADA) include: | True | 463 (77.7) | 133 (22.3) |
| Coordinating anti-doping activities worldwide | True | 393 (65.9) | 203 (34.1) |
| Maintaining the World Anti-Doping Code | True | 384 (64.4) | 212 (35.6) |
| Maintaining a list of prohibited substances and methods in sports | True | 250 (41.9) | 346 (58.1) |
| Accreditation and reaccreditation of laboratories for sample analysis | False | 353 (59.2) | 243 (40.8) |
| Prosecution of doping offenders in sports | | | |
| **Knowledge on definition of doping**<br>1. Doping violations include: | True | 471 (79.0) | 125 (21.0) |
| Attempted use of a prohibited substance or method by an athlete | True | 397 (66.6) | 199 (33.4) |
| Attempted administration of a prohibited substance or method to an athlete by an athlete support person | True | 310 (52.0) | 286 (48.0) |
| Acts by an athlete support person to discourage or retaliate against reporting to authorities | True | 332 (55.7) | 264 (44.3) |
| Failure of the athlete to submit to sample collection without compelling justification after notification | True | 335 (56.2) | 261 (43.8) |
| Possession of a prohibited substance or method by an athlete support person | | | |
| **Knowledge on roles of athlete support personnel**<br>1. The roles of athlete support person include: | True | 521 (87.4) | 75 (12.6) |
| Being knowledgeable about and to comply with all anti-doping policies and rules | True | 412 (69.1) | 184 (30.9) |
| Cooperation with the athlete-testing program | False | 198 (33.2) | 398 (66.8) |
| Being responsible for what the athlete eats and use, in the context of anti-doping | True | 302 (50.7) | 294 (49.3) |
| Using their influence on athlete values and behaviour to foster anti-doping attitudes | | | |

anti-doping without referring to materials from WADA. This study also reported that only 41.4% of the respondents had attended courses on doping in sports, and 55.5% of the respondents claimed that they update themselves on the topics of doping in sports, but only 11.8% of them do it regularly at least once a week. Table 4 summarizes the practice and experience of the respondents on doping issues.

**Table 3. The respondents' attitude towards performance enhancement (n = 596).**

| Variables | Number of respondents with negative attitude (disagree) | Number of respondents with positive attitude (agree) |
|---|---|---|
| Doping is necessary to be competitive | 462 (81.6%) | 134 (22.4%) |
| Doping is not cheating since everyone does it | 512 (85.9%) | 84 (14.1%) |
| Only the quality of performance should matter, not the way athletes achieve it | 406 (68.1%) | 190 (31.9%) |
| Athletes should not feel guilty about breaking the rules and taking performance-enhancing drugs | 501 (84.1%) | 95 (15.9%) |
| The risks related to doping are exaggerated | 451 (75.7%) | 145 (24.3%) |
| Doping is an unavoidable part of the competitive sport | 413 (69.3%) | 183 (30.7%) |
| Legalizing performance enhancements would be beneficial for sports | 428 (71.8%) | 168 (28.2%) |
| There is no difference between drugs, fiberglass poles and speedy swimsuits that are all used to enhance performance | 463 (77.7%) | 133 (22.3%) |

The study reported no significant difference between the respondents' professional status and their experience in receiving enquiries from the athletes about medication use in sports (p = 0.33) and supplements use in sports (p = 0.27). Surprisingly, there was no significant association between updating practices on doping topics and knowledge scores (t(595) = 0.79, p = 0.43). However, attendance at anti-doping courses was associated with higher knowledge scores (16.53±5.50) compared to those who had never attended such courses (15.54±5.04, t (595) = 2.28, p = 0.02). Besides, the respondents' practice of publicly support anti-doping and practice of promoting the athletes to be good role models are not significantly associated with PEAS (p = 0.84; p = 0.82).

## Discussion

This is the first multi-country study that extends our understanding of the ASP working with youth athletes from Southeast Asian countries on their current knowledge, attitudes, and experiences regarding doping in sports.

The current study revealed that most respondents correctly identified anabolic-androgenic steroids (AAS) and stimulants as prohibited substances in sports, aligning with previous surveys conducted among ASP to assess their knowledge of prohibited substances [9,26]. These substances are commonly abused by athletes, and the anti-doping testing figures report published by WADA showed that they made up almost half of the adverse analytical findings reported [27]. Conversely, caffeine, widely used by athletes as an ergogenic aid to improve endurance, muscle velocity, and strength, was correctly identified as non-prohibited by most respondents [28]. After the prohibited status of caffeine was lifted by WADA in 2004, it was found in more than 70% of the urine samples collected after competitions from 2004 to 2015, proving its prominent use and popularity among elite athletes [29].

Nevertheless, insulin, beta-blockers (or medications for high blood pressure), and beta-2 agonists (or medications for asthma) were not identified by most respondents as prohibited or potentially prohibited substances. This lack of awareness may stem from misconceptions regarding their legitimate medical use. For instance, insulin is normally used by diabetic patients to treat high sugar levels, but it could be misused by bodybuilders and weightlifters to suppress proteolysis and increase protein synthesis for faster muscle gain [30]. Meanwhile, beta-blockers are prohibited in competition for certain sports only, such as archery and golf,

**Table 4. The respondents' experience and practice regarding doping issues (n = 596).**

| Domains/ Variables | Numbers (n) | Percentage (%) |
|---|---|---|
| **Request for information from athletes** | | |
| 1. Advised athletes about anti-doping without referring materials from the World Anti-Doping Agency? | 272 | 45.6 |
| | 279 | 46.8 |
| Yes | 45 | 7.6 |
| No | 326 | 54.7 |
| Not relevant | 230 | 38.6 |
| 2. Approached by athlete for information about the proper use of medications in sports? | 40 | 6.7 |
| | 340 | 57.0 |
| Yes | 256 | 43.0 |
| No | | |
| Not relevant | | |
| 3. Asked for information about dietary supplements (including sports food and nutritional ergogenic aids) for use in enhancing sports performance? | | |
| Yes | | |
| No | | |
| **Supply of medicines/ supplements/ substances for athletes** | | |
| 1. Prescribed/dispensed or been asked to purchase any drugs/ supplements/ substances for body slimming/ muscle gain? | 87 | 14.6 |
| | 458 | 76.8 |
| Yes | 51 | 8.6 |
| No | 143 | 24.0 |
| Not relevant | 361 | 60.6 |
| 2. Suspected that the medications prescribed/ dispensed/ purchased for an athlete for the treatment of a disease were actually used to improve their sports performance? | 92 | 15.4 |
| | 236 | 39.6 |
| Yes | 360 | 60.4 |
| No | | |
| Not relevant | | |
| 3. Promoted dietary supplement (including sports food and nutritional ergogenic aids) to an athlete? | | |
| Yes | | |
| No | | |
| **Course / Training in doping in sports** | | |
| 1. Attended any courses/seminars/talks on doping in sports? | 247 | 41.4 |
| Yes | 349 | 58.6 |
| No | 331 | 55.5 |
| 2. Keep updated on the topics of doping in sports? | 265 | 44.5 |
| Yes | 18 | 5.5 |
| No | 21 | 6.4 |
| 3. How often do you update yourself on the topics of doping in sports? | 118 | 35.7 |
| More than once a week | 173 | 52.4 |
| Once a week | | |
| Once a month | | |
| Rarely | | |
| **Stance against doping in sports** | | |
| 1. Over the past 3 months, have you ever publicly claimed to support anti-doping? | 254 | 42.6 |
| Yes | 342 | 57.4 |
| No | 492 | 82.6 |
| 2. Encouraged athletes to be good role models in anti-doping? | 104 | 17.4 |
| Yes | | |
| No | | |

and less than 1% of doping tests were positive for beta-blockers, suggesting infrequent misuse by athletes [19,27]. Although beta-2 agonists are prohibited, there are a few exceptions to the commonly prescribed inhalers including salbutamol, formoterol, and salmeterol [19]. Therefore, the respondents in our study who were mainly non-healthcare professionals might have lesser awareness of the status of prohibition of these substances, which is comparable to previous findings [8–10]. Taken together, these results implied that while ASP are knowledgeable

about well-known prohibited substances, they are not conversant with doping violations associated with commonly prescribed medications.

The study also sheds light on ASP familiarity with the definition of anti-doping rule violations (ADRVs). While most respondents were aware of what constitutes ADRVs, some areas, such as possession of prohibited substances or methods by an ASP and discouraging or retaliating against the reporting to authorities, were less understood. Coaches, comprising the majority of respondents in the study, demonstrated better knowledge of ADRVs, consistent with previous findings [7]. Furthermore, the respondents exhibited awareness of the roles of WADA and were familiar with the NADO in their respective countries. This could be attributed to the fact that the respondents were mainly from international academic institutions or sports schools, and they were more likely to have experience with their NADO. Even though some of the NADOs were recently established, the efforts of the national organisations led by SEARADO to promote anti-doping were deemed significant when most of the ASP were able to identify the respective NADOs correctly. The familiarity of the ASP with their respective NADOs is essential as a core reference point should they have any doubts or uncertainties related to anti-doping.

However, respondents were not well-informed about their own roles and responsibilities in fostering anti-doping attitudes among athletes. Instead, they believed they were responsible for what the athletes eat and use, which is actually the athlete's own responsibility. The lack of awareness of their potential roles in promoting anti-doping attitudes among athletes is common among ASP, especially coaches, who prioritize training over anti-doping education in their routines with athletes, considering it more relevant in preparation for competitions [7,9]. This finding underscores the need for increased emphasis on anti-doping education among ASP.

Our study evaluated the relationship between the respondents' socio-demographic information and their knowledge scores. Previous literature suggested that the level of knowledge among ASP appeared to be related to several factors, including gender, age, experience, and professional roles [5–9]. Nonetheless, no significant differences were observed between the knowledge scores for HCP and non-HCP, contrary to previous findings [9]. This could be explained by the difference between the types of HCP recruited in our study, which mainly consisted of general practitioners, nurses, physiotherapists, and psychologists. Thus, the specific knowledge related to doping in sports may not be comparable with that of sports physicians who are specially trained in sports medicine. Likewise, no associations were found between knowledge scores and years of practice or previous athlete status. This contradicts past literature, which reported that more experienced and older coaches had better knowledge than younger ones [31]. Apart from that, we discovered that the knowledge scores of our respondents were associated with gender. This observation, however, contradicts prior research indicating that male parents exhibited significantly better knowledge about doping and its side effects than their female counterparts [5].

Despite knowledge gaps, it is reassuring that the ASP generally rejected doping and had a negative attitude towards it. A past meta-analytical review examining the predictors of doping intentions, susceptibility, and behaviour reported that doping attitudes were a significant predictor for doping susceptibility and behaviour [32]. Hence, it is essential to look into each statement from the PEAS carefully. Among the eight statements about a positive, lenient, and permissible attitude toward doping, some respondents in our study seemed to agree that the quality of performance matters more than the way athletes achieve, and doping is an unavoidable part of the competitive sport. Interestingly, both of these statements associate doping in sports with the demands of competitive sports itself, indicating a more permissive attitude towards doping among top-tier athletes [21]. We proposed that it is plausible that the

respondents felt doping could be more prevalent among elite athletes, as most cases were reported among elite athletes.

As our respondents are ASP working with young athletes, their attitudes may influence the decisions made by young athletes. It is evident from previous behavioural studies that pressure and extremely high expectations from coaches and parents can lead young athletes to adopt health-harming behaviours, including doping [5,33]. Parents and coaches, being the superiors within the sports team hierarchy, are perceived by younger athletes as the most trusted source of information and the most important decision-makers in their athletic journey [34]. In addition, if the coaching climate and team environment are generally pro-doping and lenient toward use of performance enhancers, young athletes may wind up justifying that use of performance-enhancing substances, including doping agents, as acceptable and necessary to compete on a "level playing field" [34].

We also found that ASP with greater knowledge about doping in sports tended to have a more negative attitude toward doping. Contrary to expectations, this study did not find a significant difference in attitude toward performance enhancement based on previous participation in sports as athletes. Blank et al. (2013) reported that parents who had pursued sporting careers in the past showed a significant difference in their attitudes toward doping compared to those who had not been competitive athletes [5]. As they had experienced the realities of doping during their competitive athletic journey, their acceptance of the doping phenomenon may have been different. Furthermore, gender differences in doping attitude mirrored past study by Alaranta et al. (2006), indicating that male respondents may have a more permissive attitude toward the use of doping agents [35].

The practices of ASP regarding anti-doping activities accords with our earlier observations of negative attitudes. Most respondents claimed to encourage athletes to be good role models in anti-doping, but few publicly supported anti-doping efforts. This presents an opportunity for NADOs to engage ASP in social media campaigns to create anti-doping awareness. This can be effective as younger athletes are more likely to engage in social media, and anti-doping social norms and key messages can be reinforced via these platforms [36]. On the other hand, approximately half of the respondents reported being approached by athletes seeking information about medication and supplements use in sports. Some respondents admitted to advising athletes without referring to information provided by official anti-doping agencies. This is in line with the findings of Mazanov et al. (2014) that some ASP are willing to overlook practices despite lacking professional training and sufficient knowledge to be the primary source of information on anti-doping and supplements use in sports [9]. Our study also reported that only a small number of the respondents had attended courses on doping in sports. A systematic review by Barnes et al. (2022) on coaches found that most coaches never engaged in formal anti-doping education but relied on self-education from internet resources or past experiences [7]. This is worrying, as the information provided by untrained ASP may be incorrect and, if not carefully verified, could lead to inadvertent use of prohibited substances. Moreover, most ASP in our study rarely kept themselves updated on doping in sports topics. ASP may prioritize other activities, especially training with athletes, amidst busy schedules. This is consistent with past literature where doping issues were often given low priority by athletes and support personnel [37].

This study highlighted deficiencies in anti-doping awareness, indicating areas for improvement as noted by anti-doping agencies. While most ASP in our study were non-HCPs who may not require extensive knowledge of prohibited substances in sports, it is recognized that knowledge is considered a protective factor against doping [32,38]. The WADC mandates all ASP to be knowledgeable about and comply with anti-doping policies and rules to prevent unintentional doping rule violations [1]. By identifying key gaps in understanding and

engagement with anti-doping practices among ASP in Southeast Asia, this research contributes to the existing body of knowledge on doping prevention strategies, particularly among younger athletes. Our questionnaire covered comprehensive aspects of the WADC, including overview of anti-doping initiatives, the Prohibited List, Therapeutic Use Exemption, and Athlete Biological Passport, ensuring thorough insight into anti-doping practices, which highlights the need for targeted educational interventions tailored to this population. Educational materials with learning objectives focused on remembering, understanding, and applying knowledge in daily routines can lay foundations for future training and promote critical thinking and problem-solving skills beneficial for interacting with different athlete groups [39]. Furthermore, educational efforts should underscore values and the indispensable roles ASP play in anti-doping. Anti-doping agencies should subsequently engage ASP with action-oriented programme to promote anti-doping behaviour more significantly, encouraging uptake and consistency of such behaviour.

## Limitations and conclusions

The study has several limitations that should be acknowledged. Firstly, there were varying and relatively low response rates from the participating countries, estimated at 10 to 50% of the respective populations. Subgroup analysis was not conducted in view of the significant disparity in respondent numbers across different countries. Although the sample collected was nationally representative of the ASP working with the youth athletes, the findings cannot be extrapolated to the parents of the athletes. Another limitation that we would like to acknowledge is that the self-administered questionnaire depends on the accuracy of self-reporting. While anonymity was assured, respondents may have answered in a socially desirable manner, particularly regarding their attitudes. Although socially desirable bias could be mitigated by not directly observing respondents during questionnaire completion, ASP might refer to external sources when answering the knowledge section, potentially leading to overestimation of knowledge scores and attitudes.

Notwithstanding these limitations, the study provides empirical insights into the knowledge gaps, attitudes, and experiences of ASP from Southeast Asia regarding anti-doping. While respondents were generally aware of key prohibited substances such as anabolic-androgenic steroids and stimulants, knowledge of less commonly discussed substances like insulin and beta-2 agonists was lacking. The respondents had a reasonable understanding of anti-doping violations and the roles of key organisations like WADA, but gaps remained in areas such as the athlete biological passport and ASP's roles. Despite insufficient knowledge, ASP exhibited negative attitudes towards doping in sports. In practice, over half of the respondents had experience advising athletes on medications and supplements; however, nearly half did so without referencing WADA resources. Attendance at anti-doping courses was linked to higher knowledge scores, highlighting the need for regular educational interventions.

Improving the poor to moderate levels of knowledge and attitudes among ASP could enhance and bolster the values of doping-free sports among the young athletes under their care. Enhancing ASP's understanding of anti-doping regulations and the Prohibited List would enable them to mentor young athletes more effectively, fostering integrity in sports. Given that parents were not included in the study, future research could focus on investigating the KAP of parents regarding anti-doping. Anti-doping agencies could organize outreach programme during sports competitions or school sports to provide parents with opportunities to learn about anti-doping. Cultivating positive anti-doping attitudes both at home (with parents) and in schools (with other support personnel) could be more effective in developing a strong anti-doping mindset among young athletes. With the growing emphasis on anti-doping in

induction packages for ASP, it may become easier and more acceptable for ASP to talk about anti-doping with young athletes. Considering the observed lack of self-initiative among ASP regarding anti-doping education, relevant authorities could develop policies to ensure mandatory anti-doping courses for all ASP at the school level, fostering a better understanding of anti-doping ideology and administration among ASP.

## Supporting information

**S1 Raw data.**
(XLSX)

## Acknowledgments

We would like to extend our sincere appreciation to representatives from Southeast Asia Regional Anti-Doping Organization (SEARADO), Brunei Darussalam Anti-Doping Committee (BDADC), Cambodia Anti-Doping Agency (CADA), Indonesia Anti-Doping Organisation (IADO), Lao National Anti-Doping Organisation (Lao-NADO), Anti-Doping Agency of Malaysia (ADAMAS), Myanmar Anti-Doping Organisation (MADO), Philippine National Anti-Doping Organisation (Phi-NADO), Anti-Doping Singapore (ADS), Doping Control Agency of Thailand (DCAT), Timor-Leste National Anti-Doping Organisation (Timor-leste NADO), Vietnam Anti-Doping Centre (VADC), and all the respondent who participated in our study.

Finally, we would like to express our gratitude to all the reviewers involved in reviewing the data collection form and the statistician from the Biostatistics Research Consultation Clinic, National Institute of Health Malaysia, for their invaluable help, guidance, and support throughout the project.

## Author Contributions

**Conceptualization:** Ming Chiang Lim, Gobinathan Nair, Eng Wee Chua, Tuan Mazlelaa Tuan Mahmood, Adliah Mhd Ali.

**Data curation:** Ming Chiang Lim, Gobinathan Nair.

**Formal analysis:** Ming Chiang Lim, Eng Wee Chua, Tuan Mazlelaa Tuan Mahmood, Ahmad Fuad Shamsuddin, Adliah Mhd Ali.

**Funding acquisition:** Ming Chiang Lim, Gobinathan Nair, Eng Wee Chua, Tuan Mazlelaa Tuan Mahmood, Ahmad Fuad Shamsuddin, Adliah Mhd Ali.

**Investigation:** Ming Chiang Lim.

**Methodology:** Ming Chiang Lim, Gobinathan Nair, Eng Wee Chua, Tuan Mazlelaa Tuan Mahmood, Ahmad Fuad Shamsuddin, Adliah Mhd Ali.

**Project administration:** Ming Chiang Lim, Gobinathan Nair, Eng Wee Chua, Tuan Mazlelaa Tuan Mahmood, Farrah-Hani Imran, Ahmad Fuad Shamsuddin, Adliah Mhd Ali.

**Resources:** Ming Chiang Lim, Gobinathan Nair, Farrah-Hani Imran.

**Software:** Ming Chiang Lim.

**Supervision:** Gobinathan Nair, Eng Wee Chua, Tuan Mazlelaa Tuan Mahmood, Ahmad Fuad Shamsuddin, Adliah Mhd Ali.

**Validation:** Eng Wee Chua, Tuan Mazlelaa Tuan Mahmood, Farrah-Hani Imran, Ahmad Fuad Shamsuddin, Adliah Mhd Ali.

**Visualization:** Gobinathan Nair, Eng Wee Chua, Tuan Mazlelaa Tuan Mahmood, Ahmad Fuad Shamsuddin, Adliah Mhd Ali.

**Writing – original draft:** Ming Chiang Lim.

**Writing – review & editing:** Ming Chiang Lim, Gobinathan Nair, Eng Wee Chua, Tuan Mazlelaa Tuan Mahmood, Farrah-Hani Imran, Ahmad Fuad Shamsuddin, Adliah Mhd Ali.

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
