## [Decision Letter · Decision Letter 0]

20 Sep 2024

PONE-D-24-25071Cultivating clean sport environment with athlete support personnel (ASP): A study on anti-doping knowledge, attitudes, and practices of ASPPLOS ONE

Dear Dr. Mhd Ali,

Thank you for submitting your manuscript to PLOS ONE. After careful consideration, we feel that it has merit but does not fully meet PLOS ONE’s publication criteria as it currently stands. Therefore, we invite you to submit a revised version of the manuscript that addresses the points raised during the review process.

**ACADEMIC EDITOR: **

Thank you for your submission, "Cultivating clean sport environment with athlete support personnel (ASP): A study on anti-doping knowledge, attitudes, and practices of ASP." After careful review, both reviewers recommend minor revisions to improve clarity and completeness.

**Key Revisions:**

**Data Availability**: Ensure that all data underlying your findings are clearly available as per PLOS ONE’s data policy.

**Language and Clarity**: Address grammatical issues, such as subject-verb agreement and sentence structure, and remove redundancy in the text.

**Introduction**: Strengthen the focus on the relevance of anti-doping education in Southeast Asia by providing concrete examples of its regional impact.

**Methods**: Justify the exclusion of parents from the study and clarify inclusion criteria to enhance generalizability.

**Results:** Consider adding subheadings for better organization and readability.

**Discussion and Conclusion**: Improve citation linkage, compare findings with other studies, and emphasize key conclusions for stronger impact.

We appreciate your effort and look forward to your revised manuscript.

We look forward to receiving your revised manuscript.

Kind regards,

Clementswami Sukumaran, PhD

Academic Editor

PLOS ONE

Journal Requirements:

2. Thank you for stating the following financial disclosure: Ming Chiang, Gobinathan Nair, Eng Wee, Tuan Mazlelaa, Ahmad Fuad, Adliah

Social Science Research Grant Program (2021 A-9, NF-2023-002)

World Anti-Doping Agency 

https://www.wada-ama.org/en

Approval to publish was received from the funder, no amendment was done by the funder on the study protocol and final manuscript.  

Reviewers' comments:

Reviewer's Responses to Questions

**Comments to the Author**

1. Is the manuscript technically sound, and do the data support the conclusions?

Reviewer #1: Yes

Reviewer #2: Yes

2. Has the statistical analysis been performed appropriately and rigorously? 

Reviewer #1: Yes

Reviewer #2: Yes

3. Have the authors made all data underlying the findings in their manuscript fully available?

Reviewer #1: No

Reviewer #2: No

4. Is the manuscript presented in an intelligible fashion and written in standard English?

Reviewer #1: No

Reviewer #2: Yes

5. Review Comments to the Author

Reviewer #1: Data Availability: The manuscript does not clearly indicate whether all data underlying the findings are fully available without restriction.

Language and Clarity: The language in the manuscript is generally clear, but there are several grammatical errors that need to be corrected to ensure clarity and precision.

Below are a few examples:

Subject-verb agreement issues: For example, using "is" instead of "are" when referring to plural subjects.

Redundancy: Phrases such as "Ethical approval and approval to conduct the study..." are redundant.

Missing commas: Complex sentences lack necessary commas, making the text difficult to follow.

Other Comments

Introduction: The study clearly identifies the problem as worthless knowledge and participation of Athlete Support Personnel (ASP) regarding anti-doping education. However, a specific focus is needed to address why this issue is particularly important in the context of Southeast Asia.

The study provides a comprehensive overview of the existing literature, indicating a widespread lack of knowledge and participation in anti-doping education among ASP.

The study highlights a significant gap in participation and the appropriate dissemination of information among ASPs in Southeast Asian countries regarding anti-doping education. This section could be improved by providing more concrete examples of how negative outcomes have occurred for athletes in the region.

Discussion and Conclusion:

The summary could be strengthened by more clearly stating the broader scientific implications, particularly how it contributes to the existing body of knowledge or fills a specific gap in doping research.

The research could further develop this section by explicitly comparing its findings with those from other regions or with previous global studies, thereby placing the research within a broader scientific framework.

Reviewer #2: I would like to thank you for the opportunity to review this article. The manuscript presents a relevant study on anti-doping knowledge, attitudes, and practices among Athlete Support Personnel (ASP) in Southeast Asia and addresses an important and interesting issue in the field.

I find the abstract to be mostly correct. However, I suggest revising the first part, as there appears to be duplicated information (lines 25-27).

The introduction is thorough and provides context for the research problem, as well as relevant prior literature. Nonetheless, I recommend revising the following points:

- Line 60: Please review the expression "number of literatures," as this phrasing is awkward.

- Line 82: Consider revising the phrase "However, there is no excuse to their ignorance," to avoid informal language.

The materials and methods section is solid and presents most of the necessary information for replicating the study. As a suggestion for improvement, it would be enriching to include clearer and more detailed inclusion criteria. Additionally, the exclusion of parents due to time, resource, and language limitations is mentioned, yet, given their recognized significant influence, it would be important to provide a more in-depth explanation of this decision. Consider how their exclusion might affect the results and generalizability of the study, especially since the role of parents is briefly discussed in line 368. Overall, it would be beneficial to justify this decision more thoroughly and consistently. Moreover, in line 129, it seems that the "%" symbol is missing after "60".

The results section is detailed, precise, and scientific. My only suggestion here would be to improve the structuring of the information, perhaps by using subheadings or clearly beginning each paragraph with the specific results being presented.

While the information is accurate, these suggestions would enhance readability for a potential audience. Also, in the results section, there appears to be a missing comma between lines 241 and 242.

For the discussion, I recommend improving the linkage between the text and citations, especially with previous works mentioned in the paragraph between lines 293 and 301. Additionally, the references mentioned in lines 339-341 should be explicitly cited.

In the conclusions section, I believe it would be beneficial to include clearer conclusions. While the limitations, future research directions, and practical implications are presented, I feel there is a need for stronger emphasis on the final conclusions of the study. Specifically, the key findings should be highlighted in a way that leaves the reader with clear, concise ideas at the end of the article.

I would like to congratulate the authors on their hard work and hope that my comments are helpful in improving the quality of the manuscript.

6. PLOS authors have the option to publish the peer review history of their article (what does this mean?). If published, this will include your full peer review and any attached files.

Reviewer #1: No

Reviewer #2: No

---

## [Author Response · Author response to Decision Letter 0]

28 Oct 2024

Journal Requirements:

2. Thank you for stating the following financial disclosure: Ming Chiang, Gobinathan Nair, Eng Wee, Tuan Mazlelaa, Ahmad Fuad, Adliah

Social Science Research Grant Program (2021 A-9, NF-2023-002) World Anti-Doping Agency 

https://www.wada-ama.org/en Approval to publish was received from the funder, no amendment was done by the funder on the study protocol and final manuscript. 

If this statement is not correct you must amend it as needed. Please include this amended Role of Funder statement in your cover letter; we will change the online submission form on your behalf.

The statement “The funders had no role in study design, data collection and analysis, decision to publish, or preparation of the manuscript.” was added in the cover letter as suggested. 

The statement related to ethics approval was removed from the “Acknowledgement” section as suggested. 

The raw data was attached in the Supporting Information files as suggested. 

5. Please review your reference list to ensure that it is complete and correct. If you have cited papers that have been retracted, please include the rationale for doing so in the manuscript text or remove these references and replace them with relevant current references. Any changes to the reference list should be mentioned in the rebuttal letter that accompanies your revised manuscript. If you need to cite a retracted article, indicate the article’s retracted status in the References list and also include a citation and full reference for the retraction notice.

All references were checked for their completeness and accuracy, and the reference list was updated with additional references numbered 14, 15, and 16. These references were added to further support the newly added paragraph in the Introduction section (Page 4, Line 90-96).

Rebuttal Letter (Author’s feedback)

Academic Editor Reviews:

Key Revisions:

Data Availability: Ensure that all data underlying your findings are clearly available as per PLOS ONE’s data policy. 

All data underlying the findings are fully available within the main text of the manuscript and are presented in Tables 1-4. The raw data is provided in the accompanying Excel file as Supporting Information, and it is openly accessible for review.

Language and Clarity: Address grammatical issues, such as subject-verb agreement and sentence structure, and remove redundancy in the text.

Thank you for your feedback. The grammatical issues, including subject-verb agreement, sentence structure, and redundancy have been addressed to improve clarity and precision.

Introduction: Strengthen the focus on the relevance of anti-doping education in Southeast Asia by providing concrete examples of its regional impact.

Thank you for your feedback. A short paragraph was added to emphasize the importance of the issue in the context of Southeast Asian region. 

Methods: Justify the exclusion of parents from the study and clarify inclusion criteria to enhance generalizability.

Thank you for your suggestions. The inclusion and exclusion criteria of the study have been further improved with additional explanations.

Results: Consider adding subheadings for better organization and readability.

Thank you for your suggestions. Subheadings have been added to the Methods and Results sections to improve its structure.

Discussion and Conclusion: Improve citation linkage, compare findings with other studies, and emphasize key conclusions for stronger impact.

Thank you for your suggestions. Linkage between the text and citations in the paragraph was improved by explicitly clarifying the connection between our findings and the previous works mentioned. The Conclusion section of the manuscript has been revised to emphasise the key findings, presenting them in a clearer and more comprehensive manner. 

Reviewers' comments:

Reviewer's Responses to Questions

Comments to the Author

1. Is the manuscript technically sound, and do the data support the conclusions? The manuscript must describe a technically sound piece of scientific research with data that supports the conclusions. Experiments must have been conducted rigorously, with appropriate controls, replication, and sample sizes. The conclusions must be drawn appropriately based on the data presented.

Reviewer #1: Yes

Reviewer #2: Yes

2. Has the statistical analysis been performed appropriately and rigorously?

Reviewer #1: Yes

Reviewer #2: Yes

3. Have the authors made all data underlying the findings in their manuscript fully available?

Reviewer #1: No

Reviewer #2: No

4. Is the manuscript presented in an intelligible fashion and written in standard English?

Reviewer #1: No

Reviewer #2: Yes

5. Review Comments to the Author

Reviewer #1: Data Availability: The manuscript does not clearly indicate whether all data underlying the findings are fully available without restriction. 

All data underlying the findings are fully available within the main text of the manuscript and are presented in Tables 1-4. The raw data is provided in the accompanying Excel file as Supporting Information, and it is openly accessible for review.

Language and Clarity: The language in the manuscript is generally clear, but there are several grammatical errors that need to be corrected to ensure clarity and precision. 

Below are a few examples:

Subject-verb agreement issues: For example, using "is" instead of "are" when referring to plural subjects.

Redundancy: Phrases such as "Ethical approval and approval to conduct the study..." are redundant.

Missing commas: Complex sentences lack necessary commas, making the text difficult to follow.

Thank you for your feedback. The grammatical issues, including subject-verb agreement, redundancy, and missing commas, have been addressed to improve clarity and precision.

Other Comments

Introduction: The study clearly identifies the problem as worthless knowledge and participation of Athlete Support Personnel (ASP) regarding anti-doping education. However, a specific focus is needed to address why this issue is particularly important in the context of Southeast Asia.

The study provides a comprehensive overview of the existing literature, indicating a widespread lack of knowledge and participation in anti-doping education among ASP. The study highlights a significant gap in participation and the appropriate dissemination of information among ASPs in Southeast Asian countries regarding anti-doping education. This section could be improved by providing more concrete examples of how negative outcomes have occurred for athletes in the region.

Thank you for your feedback. A short paragraph was added to emphasize the importance of the issue in the context of Southeast Asian region (Page 4, Line 90-96).

Discussion and Conclusion:

The summary could be strengthened by more clearly stating the broader scientific implications, particularly how it contributes to the existing body of knowledge or fills a specific gap in doping research.

The research could further develop this section by explicitly comparing its findings with those from other regions or with previous global studies, thereby placing the research within a broader scientific framework.

Thank you for your valuable feedback. While the original version already included comparisons with studies such as Barnes et al. (2022), Mazanov et al. (2014), and Blank et al. (2013), which reported findings from Austria, Spain, and a systematic review across 19 countries, we recognise that this may have been insufficient. In response to your suggestion, we have expanded the relevant section by incorporating additional comparisons with previous global studies (Page 20, Line 332; Page 21, Line 359). In addition, a short paragraph was added in Discussion section to emphasize how it helps to fill a research gap in doping issues (Page 24, Line 429-435).

Reviewer #2: I would like to thank you for the opportunity to review this article. The manuscript presents a relevant study on anti-doping knowledge, attitudes, and practices among Athlete Support Personnel (ASP) in Southeast Asia and addresses an important and interesting issue in the field.

I find the abstract to be mostly correct. However, I suggest revising the first part, as there appears to be duplicated information (lines 25-27).

The duplicated information in the Abstract has been removed as suggested (Page 2, Line 26).

The introduction is thorough and provides context for the research problem, as well as relevant prior literature. 

Nonetheless, I recommend revising the following points:

- Line 60: Please review the expression "number of literatures," as this phrasing is awkward.

- Line 82: Consider revising the phrase "However, there is no excuse to their ignorance," to avoid informal language.

The expression "number of literatures" has been revised to "number of studies" (Page 3, Line 59). Additionally, the phrase "there is no excuse to their ignorance" has been changed to "ignorance cannot be an excuse" (Page 4, Line 81).

The materials and methods section is solid and presents most of the necessary information for replicating the study. As a suggestion for improvement, it would be enriching to include clearer and more detailed inclusion criteria. Additionally, the exclusion of parents due to time, resource, and language limitations is mentioned, yet, given their recognized significant influence, it would be important to provide a more in-depth explanation of this decision. Consider how their exclusion might affect the results and generalizability of the study, especially since the role of parents is briefly discussed in line 368. Overall, it would be beneficial to justify this decision more thoroughly and consistently. Moreover, in line 129, it seems that the "%" symbol is missing after "60".

Thank you for your suggestions. The inclusion and exclusion criteria of the study have been further improved with additional explanations (Page 5, Line 116-119; Page 6, Line 121-125). Additionally, the "%" symbol has been added after "60" as suggested (Page 7, Line 141).

The results section is detailed, precise, and scientific. My only suggestion here would be to improve the structuring of the information, perhaps by using subheadings or clearly beginning each paragraph with the specific results being presented.

While the information is accurate, these suggestions would enhance readability for a potential audience. Also, in the results section, there appears to be a missing comma between lines 241 and 242.

Thank you for your suggestions. Subheadings have been added to the Results section to improve its structure. Additionally, punctuation has been added to the sentence as suggested.

For the discussion, I recommend improving the linkage between the text and citations, especially with previous works mentioned in the paragraph between lines 293 and 301. Additionally, the references mentioned in lines 339-341 should be explicitly cited.

Thank you for your suggestions. Linkage between the text and citations in the paragraph was improved by explicitly clarifying the connection between our findings and the previous works mentioned (Original version at Page 19, Line 293-301; Revised version at Page 19, Line 310-319). References have been added to the texts as suggested. (Page 21, Line 359).

In the conclusions section, I believe it would be beneficial to include clearer conclusions. While the limitations, future research directions, and practical implications are presented, I feel there is a need for stronger emphasis on the final conclusions of the study. Specifically, the key findings should be highlighted in a way that leaves the reader with clear, concise ideas at the end of the article.

The Conclusion section of the manuscript has been revised to emphasise the key findings, presenting them in a clearer and more comprehensive manner. (Page 25, Line 458-467; Line 470-471; Line 475-477).

---

## [Decision Letter · Decision Letter 1]

15 Nov 2024

Cultivating clean sport environment with athlete support personnel (ASP): A study on anti-doping knowledge, attitudes, and practices of ASP

PONE-D-24-25071R1

Dear Dr. Mhd Ali,

We’re pleased to inform you that your manuscript has been judged scientifically suitable for publication and will be formally accepted for publication once it meets all outstanding technical requirements.

Kind regards,

Clementswami Sukumaran, PhD

Academic Editor

PLOS ONE

Additional Editor Comments (optional):

Reviewers' comments:

Reviewer's Responses to Questions

**Comments to the Author**

1. If the authors have adequately addressed your comments raised in a previous round of review and you feel that this manuscript is now acceptable for publication, you may indicate that here to bypass the “Comments to the Author” section, enter your conflict of interest statement in the “Confidential to Editor” section, and submit your "Accept" recommendation.

Reviewer #1: All comments have been addressed

Reviewer #2: All comments have been addressed

2. Is the manuscript technically sound, and do the data support the conclusions?

Reviewer #1: Yes

Reviewer #2: Yes

3. Has the statistical analysis been performed appropriately and rigorously? 

Reviewer #1: Yes

Reviewer #2: Yes

4. Have the authors made all data underlying the findings in their manuscript fully available?

Reviewer #1: Yes

Reviewer #2: Yes

5. Is the manuscript presented in an intelligible fashion and written in standard English?

Reviewer #1: Yes

Reviewer #2: Yes

6. Review Comments to the Author

Reviewer #1: Upon reviewing the revised manuscript, I confirm that the authors have sufficiently addressed the critical points from the prior review.

Reviewer #2: Dear Authors,

Thank you for the revised version of your manuscript and for your responses. I appreciate the attention given to the previous feedback. I feel the main points have effectively been addressed. Overall, I consider the manuscript is now better structured and clearer.

Best regards.

7. PLOS authors have the option to publish the peer review history of their article (what does this mean?). If published, this will include your full peer review and any attached files.

Reviewer #1: No

Reviewer #2: No

---

## [Editor Report · Acceptance letter]

22 Nov 2024

PONE-D-24-25071R1 

PLOS ONE

Dear Dr. Mhd Ali, 

I'm pleased to inform you that your manuscript has been deemed suitable for publication in PLOS ONE. Congratulations! Your manuscript is now being handed over to our production team.

Kind regards, 

on behalf of

Dr. Clementswami Sukumaran 

Academic Editor

PLOS ONE